# Cannabidiol Selectively Binds to the Voltage-Gated Sodium Channel Na_v_1.4 in Its Slow-Inactivated State and Inhibits Sodium Current

**DOI:** 10.3390/biomedicines9091141

**Published:** 2021-09-02

**Authors:** Chiung-Wei Huang, Pi-Chen Lin, Jian-Lin Chen, Ming-Jen Lee

**Affiliations:** 1Department of Post Baccalaureate Medicine, Kaohsiung Medical University, Kaohsiung 80708, Taiwan; g10054b@kimo.com; 2Department of Physiology, Kaohsiung Medical University, Kaohsiung 80708, Taiwan; 3Department of Internal Medicine, Division of Endocrinology and Metabolism, Kaohsiung Medical University Hospital, Kaohsiung 80708, Taiwan; pichli@kmu.edu.tw; 4Department of Physiology, National Taiwan University College of Medicine, Taipei 100233, Taiwan; r07441009@g.ntu.edu.tw; 5Department of Neurology, National Taiwan University Hospital, Taipei 100233, Taiwan

**Keywords:** cannabidiol, Na_v_1.4 channel, myotonia, fast inactivation, slow-inactivation

## Abstract

Cannabidiol (CBD), one of the cannabinoids from the cannabis plant, can relieve the myotonia resulting from sodium channelopathy, which manifests as repetitive discharges of muscle membrane. We investigated the binding kinetics of CBD to Na_v_1.4 channels on the muscle membrane. The binding affinity of CBD to the channel was evaluated using whole-cell recording. The CDOCKER program was employed to model CBD docking onto the Na_v_1.4 channel to determine its binding sites. Our results revealed no differential inhibition of sodium current by CBD when the channels were in activation or fast inactivation status. However, differential inhibition was observed with a dose-dependent manner after a prolonged period of depolarization, leaving the channel in a slow-inactivated state. Moreover, CBD binds selectively to the slow-inactivated state with a significantly faster binding kinetics (>64,000 M^−1^ s^−1^) and a higher affinity (K_d_ of fast inactivation vs. slow-inactivation: >117.42 μM vs. 51.48 μM), compared to the fast inactivation state. Five proposed CBD binding sites in a bundle crossing region of the Na_v_1.4 channels pore was identified as Val793, Leu794, Phe797, and Cys759 in domain I/S6, and Ile1279 in domain II/S6. Our findings imply that CBD favorably binds to the Na_v_1.4 channel in its slow-inactivated state.

## 1. Introduction

The voltage-gated sodium channel is a hetero-multimeric protein composed of a large ion-conducting, voltage-sensing α-subunit and a few smaller β-subunits [1,2,3,4,5]. The voltage-gated sodium channel Na_v_1.4 of the skeletal muscle is a heterodimer consisting of a pore-forming α and a regulatory β1-4 subunit [6,7]. The α-subunit contains four homologous domains (DI–DIV), each with six transmembrane segments [6,7]. During depolarization, the S4 segments in each domain, containing positive amino acid residues and functioning as voltage sensors, can move outward to alter the channel conformation and modify its biophysical property [1,2,3,4,5,6,7]. The different charge residues of the S4 segments play an important role in determining domain-specific functions [6,7]. The S4 segments in the domains DI and DII are thought to be key molecules for the activation of the sodium channel, whereas the S4s in DIII and DIV regulate the fast inactivation process. The pore with its selectivity filter is lined by the loops formed between the S5 and S6 segments of each domain [1,2,3,4,5]. It has been reported that the pore of the Na_v_1.4 channel can be an interaction site for pharmacological blockers, such as lidocaine, benzocaine, and ranolazine [8,9]. The pore exhibits four intra-bilayer fenestrations, connecting it with the bilayer core. Nevertheless, their functional role remains elusive [10].

Most of the skeletal muscle channelopathies arise from *de novo* or autosomal dominant mutations in the *SCN4A* gene, which encodes the α-subunit of the Na_v_1.4 channel [11,12,13,14]. The mutations in the Na_v_1.4 channel result in alterations of skeletal muscle excitability [15,16,17]. Non-dystrophic myotonia with increased excitability of the muscle membrane, such as myotonia congenita (MC) and paramyotonia congenita (PMC), are the primary clinical phenotypes [15,17]. The gain-of-function mutations in MC and PMC enhance inward Na^+^ currents, leading to an impairment of activation with a hyperpolarized shift [11,15,18,19,20]. The study of these disorders has provided insights into the structure–function relationship of the sodium channel and rational approaches for therapeutic intervention in many disorders associated with cellular hyperexcitability [21]. Hyperexcitable muscle channelopathies with mutations in the Na_v_1.4 channel have been classified into non-dystrophic myotonia and periodic paralyzes [11,20]. Electrophysiological studies in a heterologous expression system with Na_v_1.4 mutations have shown variable biophysical defects in activation, fast inactivation, and slow-inactivation [22,23]. The mutations could lead to an initial myotonia discharge burst and result in stiffness and weakness in affected patients [24,25].

Recently, cannabidiol (CBD) was reported to be a potentially beneficial compound to treat epilepsy in children [26,27]. CBD is one of the 113 identified cannabinoids extracted from the cannabis plant [28]. Trials using CBD to treat a few disease entities such as anxiety, cognition decline, movement disorders, and pain have been conducted, although high-quality evidences to prove the effectiveness have not yet been obtained [29,30]. In addition, the CBD-enriched hemp oil was found to have the anti-cancer property with increasing of oxidative stress [31]. There are a few routes of CBD ingestion, it can be taken directly by mouth, the cannabis smoke can be inhaled by nose, or the aerosol spray of CBD can be absorbed by the cheek mucosa [32,33].

Moreover, CBD was suggested to be a potentially therapeutic compound against a variety of conditions, such as muscle spasms, pain, and myotonia [11]. Some reports of CBD’s efficacy are anecdotal, whereas others were experimentally and clinically substantiated [34,35,36]. CBD showed therapeutic efficacy against Dravet and Lennox Gastaut syndromes in clinical trials phase III [11,34,35,36]. The efficacious dosage against the Dravet syndrome is around 20 mg/kg [35]. However, the mechanism underlying CDB efficacy and its associated biophysical changes on channelopathies remain elusive. Considering the lack of therapy for myotonia resulting from Na_v_1.4 channel mutations, we investigated whether CBD could represent another drug of choice for the treatment of symptoms.

Our results suggest that CBD can selectively bind to the Na_v_1.4 channel in its inactivated state with a dissociated constant of approximately 31.5 μM. Moreover, the binding affinity for the fast-inactivated state was four times lower than that for the slow-inactivated one. CBD exhibited significantly higher binding kinetics to the slow—compared to the fast—inactivated Na_v_1.4 channels. Using the molecular docking model, we identified five CBD binding sites at the bundle crossing region in the Na_v_1.4 channel pore, which may affect conformational changes in the slow-inactivated gate. These results suggest that CBD could serve as a distinct therapeutic drug for Na_v_1.4 channelopathies.

## 2. Experimental Section

### 2.1. Wild-Type (WT) Na_v_1.4 Channel CDNA Constructs

The cDNA clone with the *SCN4A* gene, which encodes the α-subunit of sodium channel Na_v_1.4, was obtained from OriGenes Technologies Company (Cat. No. RC218290; Rockville, MD, USA). The SCN4A cDNA was seamlessly cloned into the mammalian expression cloning vector pcDNA3.1(+)-DYK vector [37,38].

### 2.2. Transient Transfection of Chinese Hamster Ovary (CHO-K1) Cells

The cells were grown and incubated under the culture condition as in our previous study [37,38]. Briefly, CHO-K1 cells were incubated in the F12K medium (Thermo Fisher Scientific, Waltham, MA, USA) with 5% CO_2_ at 37 °C, supplemented with 10% fetal bovine serum (Thermo Fisher Scientific) and 0.2% Normocin (InvivoGen, San Diego, CA, USA). The cells with a density of 1.0 × 10^6^ were seeded onto a 3.5 cm culture dish (Greenpia Technology, Seoul, South Korea). Transient transfection was carried out using the Lipofectamine™ 3000 reagent (Thermo Fisher Scientific). After 24 h of transfection, the cells were washed with the F12K medium. Then, the CHO-K1 cells stayed in culture for 96 h until the electrophysiological experiments.

### 2.3. Whole-Cell Patch-Clamp Recording

Before the whole-cell patch-clamp recording, the proteinases IIX-III (0.025 mg/mL; Sigma–Aldrich, St. Louis, MO, USA) were added to the transfected cells. CHO-K1 cells with the expressed Na_v_1.4 channel were plated onto 1.2 cm coverslips (Paul Marienfeld GmbH, Lauda-Königshofen, Germany) at 37 °C for approximately 45 min. Whole-cell patch-clamp recordings were carried out within four days of transfection. Currents were recorded using an Axopatch 700B amplifier at 25 °C (Axon Instruments, Sunnyvale, CA, USA), interfaced with the pClamp 9.2 acquisition software (Molecular Devices, San Jose, CA, USA). Currents were filtered at 5 kHz with a four-pore Bessel filter and digitized at 50 μs intervals using a signal conditioning amplifier, the Digidata-1322A interface (Axon Instruments, Union City, CA, USA). For patch-clamp recording, fire-polished, borosilicate glass-pulled micropipettes were prepared using a Sutter P-97 puller (Sutter Instrument Company, Novato, CA, USA). The pipette resistance was 1.5–2.5 mΩ. The glass electrode pipette was filled with an internal solution containing the following (in mm): 75 CsCl, 75 CsF, 5 HEPES, 2 CaCl_2_, and 2.5 EGTA (pH 7.6). The whole-cell configuration was immersed in the external solution which contains the following ingredient (mm): 150 NaCl, 10 HEPES, 2 CaCl_2_, and 2.5 MgCl_2_ (pH 7.6). Electrophysiological recording started approximately 5 min after establishment of a whole-cell configuration, which allowed reaching an equilibrium with internal solution in the intracellular compartment at the imposed holding membrane voltage (especially at −120 mV).

### 2.4. Chemical Drugs

CBD was purchased from Sigma–Aldrich (Burlington, MA, USA). CBD was dissolved in methanol to a stock concentration of approximately 31.8 mm. After being aliquoted, it was stored at −20 °C. A fresh aliquot of the drug was used for each experiment, which was diluted with an external solution to reach the required concentration.

### 2.5. Molecular Docking of CBD to the Na_v_1.4 Channel

The CDOCKER program in the DS 2020 software (BIOVIA, Dassault Systems, Discovery Studio, 2020, San Diego, CA, USA) was employed to model the CBD docking to the human Na_v_1.4 channel. The molecular mimicry screened for the CBD binding site potential in the human Na_v_1.4 channel [39]. The X-ray diffraction structure of the human Na_v_1.4 channel in complex with the β-subunit was downloaded from the Protein Data Bank (PDB ID: 6GAF; https://www.ncbi.org/) (19 October 2018) [10]. Ligands were prepared to obtain adequate biological ionization and tautomerization status. The central domain of the human Na_v_1.4 channel was defined as the binding pocket. The model yielded the 10 best binding poses of CBD, ranked by the mean energy score. CDOCKER energy values of CBD with the human Na_v_1.4 channel were obtained as standard to select good predictions of affinity peptides (calculated in kcal/mol).

### 2.6. Data Analysis

All electrophysiological sweeps were analyzed using the software Clampfit 9.0 (Axon Instrument, San Jose, CA, USA). Steady-state inactivation curves were fit to a Boltzmann function to obtain the midpoint (V_h_) and slope (*k*) values. Time constants for recovery were obtained by fitting data from each cell to a first-order exponential function and the averaging time constants across cells. Statistical data were analyzed using Sigmaplot software 10.0 (Systat Software Inc, Chicago, IL, USA) and are described as means ± standard error of the mean. Data were assessed and analyzed using Student’s independent *t*-tests, and statistical significance was defined as *p* < 0.05.

## 3. Results

### 3.1. The Assessment of Transient Sodium Current Inhibition by CBD with Variable Holding Potentials on the CHO Cells Expressing Na_v_1.4 Channel

We first assessed possible changes in Na_v_1.4 channel activation after the binding of CBD. We demonstrated a negligible inhibitory effect in the activation of transient Na_v_1.4 currents in CHO-K1 cells held at −120 mV upon treatment with different concentrations (between 3 and 100 μM) of CBD (left panel of Figure 1A). However, the inhibitory effect of CBD escalated in a dose-dependent manner, while the holding potentials increased from −120 to −80 mV (Figure 1A). To confirm the influence of holding potential on the inhibitory effect of CBD, the CHO-K1 cells were next subjected to a holding potential between −70 and −120 mV, and the inhibition of sodium currents was evaluated with different concentrations of CBD (3, 10, 30, to 100 μM; Figure 1B). Compared to the depolarization currents in the control condition (no CBD), there was a significant decrease in relative currents along with the increased concentration of CBD. Moreover, the inhibitory effect was found to be enhanced with the escalation of holding potential (from −120 to −70 mV; Figure 1B). Each of the data could be well fitted by the one-to-one binding curve, which provides a rough estimate of apparent dissociation constants (K_app_) of 1341.7, 275.5, 96.9, 30.1, 10.1, and 4.1 μM for CBD binding at the holding potentials of −120, −110, −100, −90, −80, and −70 mV, respectively. The holding potential-dependent K_app_ value also reduced with escalation of the holding potential (Figure 1B). The suppression was significant at −80 mV and even more pronounced at −70 mV. Cells holding at −120 mV were nearly devoid of suppression, with only a “closed”-state Na_v_1.4 channel available. The stronger membrane depolarization was, the more abundant were the inactivated channels. These findings indicate that CBD presumably binds more favorably to the inactivated than to the resting channels. Moreover, the K_app_ values from 1341 to 4.1 μM at different holding potentials did not show a monotonous or commensurate trend of alteration along with the different concentrations of CBD, which implies that at different holding potentials, CBD binds to a mixture of variable gating states of Na_v_1.4 channels.

### 3.2. The Influence of CBD on the Inactivated Curve of Na_v_1.4 Channels Following a Short Depolarizing Pulse

Previous studies on the gating properties of sodium channels have demonstrated their conversion from open, inactivated states to closed states [40]. The inactivated channels can be obtained and classified into fast and slow-inactivation according to their period of depolarization [41]. To discern to which of the inactivated channel states CBD was preferentially bound, we examined the influence of CBD on the inactivated curve of Na_v_1.4 channels following a short depolarizing pulse (approximately 100 ms). For the fast-inactivating state, a series of depolarization prepulses, which escalated from −120 to −20 mV for 100 ms, were applied to the CHO-K1 cells, followed by a test pulse at 0 mV for 3 ms (left panel of Figure 2A). The sweep traces were recorded in both the control and CBS treatment (30 µM) conditions (right panel of Figure 2A). We defined “fraction available” as the recovered channels from fast inactivation. Then, the inactivation curve was depicted using the “fraction available” against the depolarizing membrane potential, between −120 and −20 mV (Figure 2B). The fast inactivation curve showed no significant shift upon treatment with 30–100 μM of CBD, as compared to controls (Figure 2B). Furthermore, the inactivation curve of those channels after washing out of the CBD did not show a significant deviation from the control and CBD treatment conditions (Figure 2C). Figure 2D shows no significant change in depolarizing membrane potential (ΔV) in those with 30 μM of CBD and those after washing out of CBD. The ΔV was defined as the relative depolarizing membrane potential at which 50% of the channels were available (50% of fraction available), compared to the controls. Similarly, there was no significant change in the slope factor for the inactivation curves (Figure 2D). These data indicate that steady-state occupancy of the Na_v_1.4 channel binding sites by CBD was rarely achieved in accordance with the short depolarization (fast inactivation) pulse.

### 3.3. Exploration of Whether CBD Can Bind to the Slow-Inactivated Na_v_1.4 Channel with an Elongation of the Depolarizing Prepulse

To explore whether CBD can bind to the slow-inactivated Na_v_1.4 channel, a new protocol with elongation of the depolarizing prepulse up to 18 s was designed (Figure 3A). The sweep traces at different depolarizing potentials were recorded in the control and CBD treatment (30 µM) conditions (Figure 3A). After treatment, the activation curve was inhibited as compared to the controls. The different concentrations of 3, 10, 30, and 100 μM of CBD were applied to the CHO-K1 cells, and then the availability of sodium channel (fraction available) against the escalating depolarizing prepulse potentials from −120 to −40 mV was depicted in those cells with the treatment of CBD (Figure 3B). The curve was significantly left-shifted along with the increase of CBD concentration in a dose-dependent manner. Moreover, such a shift was reversed to the original state when CBD was washed out (Figure 3C). The prominent shift of the curve suggests a variable binding affinity between CBD and the different gate states of Na_v_1.4 channels. The slope factors for the curve at different concentrations of CBD were not different from those of the controls (Figure 3D). Nevertheless, a significant increase in ΔV (the differential membrane potential between the controls and those cells with CBD treatment at 50% fraction available) was observed after treatment, in a dose-dependent manner (Figure 3E). The hyperpolarizing shift (left shift) indicates the inactivation occurs at a lower membrane potential than the controls. The larger the value of ΔV, the more efficient it is for the channel to evolve into an inactivated state. Thus, if CBD binds to the inactivated state of the Na_v_14 channel, a higher concentration of CBD would render a larger value of ΔV. According to the inactivation curve shifts (ΔV) with different concentrations of CBD, the deduced K_d_ was approximately 1.2 μM (Figure 3F) [41,42,43]. The 18 s of inactivating pulse can not only allow more time for CBD binding but also distributes the channel to different gating states beyond its fast inactivation. These findings suggest that CBD can bind to the slow-inactivated Na_v_1.4 channel after the application of variable depolarization voltages.

### 3.4. Determination of Na_v_1.4 Channel Slow Inactivated State

To evaluate the slow-inactivation of the Na_v_1.4 channel, another approach resulting in more accurate measurement was next employed. The protocol was designed as follows. After a depolarization at 0 mV for 20 ms, the time for resting membrane potential, −120 mV was prolonged successively until the next test potential, 0 mV for 20 ms (Figure 4A). Most of the inactivated Na_v_1.4 channels could recover within approximately 30 ms at −120 mV (right panel of the Figure 4A). Based on the findings, we assumed that the contribution of the available channels after fast inactivation can be minimized when the time laps was protracted longer than 30 ms. We then evaluated whether the voltage of depolarized membrane potential and the periods of the activation time (30 ms) influenced the slow-inactivation. With two different depolarizing potentials, −80 (Figure 4B) and −10 mV (Figure 4C), the relative currents that represented the available channels after slow-inactivation were evaluated. Less than 10% of the Na_v_1.4 channels entered the slow-inactivated state with a membrane potential of −80 mV in 14 s (Figure 4B), whereas approximately 90% of the Na_v_1.4 channels entered the slow-inactivated state at −10 mV (with a time constant of approximately 1.5 s, Figure 4C). Finally, we assessed the difference in the transient inactivation and the slow-inactivation curves of the Na_v_1.4 channel. To depict the slow-inactivation curve, the depolarizing membrane potential was −10 mV for 30 s and then repolarized in a 20 ms gap followed by a test pulse at 0 mV for 10 ms. The slow-inactivation curve showed a depolarizing shift compared to the fast inactivation curve (right panel of Figure 4D).

### 3.5. Evaluation of Different Depolarizing Potentials Influence on the CBD Inhibitory Effect

To further confirm that CBD can bind to the Na_v_1.4 channel in its slow-inactivation state, the channel was depolarized to −10 or −80 mV for 18 s with the treatment of different concentrations of CBD (Figure 5A,B). The available Na_v_1.4 channels were assessed by the period of time (Δt) at −120 mV, evaluating the relative currents after prolonged depolarization with a subsequent certain period of repolarization. During the period, the Na_v_1.4 channels were supposed to be recovered from the inactivated state, evolving to a closed state, which can then be activated by the test pulse. Thus, the relative currents represented the channels recovered from the inactivated state. Since the period is scaled in seconds, we assumed that only the slow-inactivated state Na_v_1.4 channels were available. Furthermore, the relative currents recorded from CHO-K1 cells under the treatment of different concentrations of CBD were the indices for the binding of CBD to the slow-inactivated Na_v_1.4 channels. The larger the relative current was, the more channels were available, which indicates a more efficient dissociation of the complex of CBD/slow-inactivated Na_v_1.4 channels (Figure 5A,B). As time (Δt) elapsed, the relative current increased quickly and attained ~90% of the controls (Figure 5A,B). With the treatment of CBD, there was a decremental change in the relative currents in a dose-dependent manner (Figure 5A,B). The slowing of the recovery kinetics was embodied by the decrease in the area under the recovery time course (Appendix A), which was already manifested by the cells treated with 3–10 μM of CBD at a depolarized pulse of −10 mV (Figure 5A,B). The effect was saturating with 100–300 μM of CBD, which was well described by a one-to-one binding curve with a K_d_ to the slow-inactivated state of the Na_v_1.4 channel at approximately 31.5 μM (Figure 5C). Since the slow-inactivated state is the dominant one after strongly depolarizing the membrane potential for such a long time, we postulated that CBD selectively binds to the slow-inactivated Na_v_1.4 channels with a K_d_ of approximately 31.5 μM. In contrast, a similar approach with an activating pulse of −80 mV for 18 s resulted in a K_d_ of 127.3 μM (Figure 5D) [41]. Since the depolarizing potential at −80 mV may not provide adequate available slow-inactivating channels, the value of K_d_ was much larger than for the channels with prepulse depolarization at −10 mV (approximately 31.5 μM). We therefore conclude that CBD can selectively bind to the slow-inactivated channel.

### 3.6. Evaluation of CBD Inhibition by the Time Elapse for Depolarization in Na_v_1.4 Channel

To further investigate the binding kinetics between Na_v_1.4 channels and CBD, the protocol illustrated in Figure 6A was designed. The Na_v_1.4 channels were depolarized at −10 mV for a variable of time period (Δt), which allowed the binding between CBD (3–100 µM) and the Na_v_1.4 channels. After a period of repolarization, −120 mV for 200 ms to eliminate the fast-inactivated Na_v_1.4 channels, a test pulse was applied to obtain the current that represented the available Na_v_1.4 channels recovered from the slow-inactivation. As shown in the lower panel of Figure 6A, the higher the concentration of CBD was, the less sodium current was obtained. The differential currents, defined as the currents from CHO-K1 cells treated with CBD at a certain time of depolarization relative to the controls, increased in a dose-dependent manner (Figure 6B). The kinetics of current decreases were found to be linearly correlated to CBD concentrations, giving a binding rate constant of approximately 64,000 M^−1^ s^−1^ (K_on_) with the treatment of 3–100 μM (Figure 6C, solid line). The unbinding rate, or the y-intercept, of the linear fit to the fastest macroscopic binding rate (i.e., approximately 64,000 M^−1^ s^−1^) in Figure 6C, was approximately 0.7 s^−1^ (K_off_). These kinetic data thus implicate a dissociation constant (K_d_ = K_off_/K_on_) of approximately 10.7 μM. These findings suggest that CBD favors binding to the Na_v_1.4 channel in its slow-inactivated state.

### 3.7. Molecular Mechanism of the Na_v_1.4 Channel with Cannabidiol

To inspect the CBD binding site on the Na_v_1.4 channel, the molecular docking model was used (Figure 7A,B) [39]. The binding sites of CBD were found to be close to domains I (green) and II (dark red) of the Na_v_1.4 channel (Figure 7C,D). The binding of domain I (DI/S6) to CBD stabilized four conventional π alkyl interactions with the residues C759, V793, L794, and F797 at the gating hinge of the bundle crossing region of DI/S6 in the Na_v_1.4 channel pore (Figure 7E). Moreover, the atom of CBD also formed one conventional π alkyl interaction with residue I1279 at the bundle crossing region DII/S6 in the Na_v_1.4 channel pore (Figure 7E,F). While considering the binding affinity between CBD and the Na_v_1.4 channel, we proposed that these five binding sites probably acted as a slow-inactivation gate in the Na_v_1.4 channel.

## 4. Discussion

### 4.1. The Binding Kinetic Analysis of Cannabidiol with Na_v_1.4 Channel

In summary, the current study designed a few electrophysiological protocols to explore the binding kinetics of CBD to the Na_v_1.4 channel. First, we excluded the binding of CBD on the open state of the gate, considering that the inhibition of the activation current increased more significantly in cells with a higher resting membrane potential (−80 mV) compared to those with a low one (−120 mV). The K_app_ at −120 mV was 1341.7 μM, which is rather different from 4.1 μM at −70 mV. These findings suggest that CBD may bind to the inactivated state of the channel. As we know, two inactivated states of the voltage-gated sodium channel, fast and slow-inactivation, can be assessed according to the time lapse after depolarization. By the short period of depolarization (100 ms), the inhibition of sodium current with different concentrations of CBD did not show any significant difference (Figure 2B). Likewise, there were no disparate findings between those with or without CBD treatment (Figure 2C). These findings concordantly suggest that the fast-inactivating gate of the channel might not be the main binding site. Using the prolonged period of depolarization (18 s), we evaluate whether CBD can bind to the slow-inactivated Na_v_1.4 channel (Figure 3). The inhibitory effect of CBD showed a marked left shift of the curve when the fraction of available channels was depicted against the different membrane potentials (Figure 3B). The higher the concentration of CBD, the more pronounced the shift. Furthermore, in addition to the left shift, the curve resumed to its original state when CBD was washed out (Figure 3C). In different concentrations, the curve with exp(ΔV/k) against the concentration of CBD fit in a straight line using a one-to-one binding model. To further validate that CBD can bind to the slow-inactivation state of the sodium channel, the current representing the available channels recovered from slow-inactivation (the time scale for depolarization is in seconds; Figure 6A) was evaluated with different concentrations of CBD. Similar to the previous findings, the current decreased along with the increased concentration (Figure 6A). Furthermore, the significant inhibition can be observed as the time for depolarization protracted longer than 1 s, when, mainly, the Na_v_1.4 channel was in the slow-inactivated state. Altogether, our findings reveal that CBD may bind to the Na_v_1.4 channel in its slow-inactivated gating state. The molecular modeling suggests that five CBD binding sites on the gating hinge of the bundling crossing region in the Na_v_1.4 channel pore were located in domain I/S6 (residues Val793, Leu794, Phe797, and Cys759) and one site of the bundling crossing region was in domain II/S6 (residue Ile1279) (Figure 7).

CBD has an inhibitory effect on the Na_v_1.4 channel of skeletal muscles [13,39]. The IC_50_ is approximately 1.9–3.8 μM, which is in the therapeutic range [13]. A recent study indicates that CBD is a compound partitioning in lipid membranes, which indirectly alters membrane fluidity and affects voltage-gated sodium channel gating conformation [13]. Moreover, CBD was found to play a role as an open channel pore blocker in voltage-gated sodium channels and has indirect effects that are involved in modulating cell membrane elasticity [11,13]. Nevertheless, these studies did not describe how CBD inhibits the sodium current of the Na_v_1.4 channel in different binding affinities and kinetics at the inactivated gate states. Herein, we provide evidence that CBD could bind to the slow-inactivated Na_v_1.4 channels and the binding is selective to the slow-inactivated state with the significantly faster binding kinetics and a much higher affinity (Figure 3, Figure 4, Figure 5 and Figure 6). These properties raise the intriguing possibility that CBD may have a therapeutic application for sodium channelopathies in muscles.

### 4.2. Clinical Implications

Our study showed that CBD binds more selectively to the slow-inactivated state with a fast-binding kinetics (>64,000 M^−1^ s^−1^) and a high affinity (K_d_ of slow-inactivation is approximately 31.5 μM) (Figure 6). A previous study showed that CBD blocks the sodium current of the Na_v_1.2 channel depending on the temperature. The lower the temperature, the stronger the potency of the inhibition [11]. In the brain slice study, a voltage-gated sodium channel inhibitor, eslicarbazepine (ESL), enhanced the slow-inactivation of voltage-gated sodium channels [45], considering the relatively slow development ESL effect. Notably, the binding kinetics of CBD to the slow-inactivated state Na_v_1.4 channel was slower than those for the classic Na^+^ channel-inhibiting anticonvulsants diphenylhydantoin (DPH), carbamazepine (CBZ), and lamotrigine (LTG), which selectively bind to the fast-inactivated state with fast-binding kinetics [42,43,46]. DPH, CBZ, and LTG bind to a common blocking site with the two characteristic aromatic rings separated by approximately 4–5 Å [46]. Oxcarbazepine and ESL are members of the dibenzoazepine family containing the diphenyl motif [47,48]. In contrast, CBD is structurally distinct from the diphenyl compounds and contains a cyclohexene and resorcin group with a major side chain [49]. CBD contains adjacent cyclohexene and resorcinol motifs [49]. The rigid 3D structure of CBD is proposed to be more stereoselective. Molecular docking modeling revealed that CBD acts as a physical pore blocker in voltage-gated Na_v_1.4 channels (Figure 7). The modeling predicted that five binding sites of CBD may act as part of the slow-inactivation gate (located in the residues of DI/S6 and DII/S6) (Figure 7). The fast-binding kinetics of CBD to the slow-inactivated Na_v_1.4 channel is different from that of the DPH since it has a much slower binding kinetics. Moreover, the K_d_ for the binding is approximately 31.5 μM, which is higher than the therapeutic concentrations (approximately 20 μM in the central nerve system) of CBD [50,51]. The findings suggest that CBD harbors a more favorable property in clinical settings compared to DPH, CBZ, and LTG [42,43,46]. If the voltage-gated Na_v_1.4 channel in the skeletal muscle had differential affinities to CBD, DPH, and CBZ, as that found in the neurons, the strong and steady-state inhibition of the Na_v_1.4 channel by CBD may suggest a superiority of CBD to DPH and CBZ for the treatment of myotonia. Because the abnormal depolarization of the affected skeletal muscle cells is possibly longer than a few seconds and the repetitive discharges may protract longer than 100 ms, CBD, which favors binding of slow-inactivated sodium channels, may represent a better choice than DPH, CBZ, and LTG in muscle channelopathies caused by a dysfunctional Na_v_1.4 channel.

According to our findings, we demonstrated that the binding affinity between CBD and the fast-inactivated Na_v_1.4 channel is extremely low, since the K_d_ (K_off_/K_on_) of fast inactivation is greater than 117.42 μM (Figure 5). CBD binds more selectively to the slow-inactivated Na_v_1.4 channel with a faster binding rate (>64,000 M^−1^ s^−1^) and a lower K_d_, of approximately 31.5 μM, which is above the highest therapeutic concentration range. The K_d_ of the binding between the slow-inactivation sodium channel and DPH is approximately 14 μM, which is below the therapeutic concentrations [52]. Therefore, CBD may be superior to DPH, while the myotonia discharges are characterized by more positive membrane voltages during the burst intervals, driving more Na_v_1.4 channels into the slow-inactivated state. In addition, physiological burst discharges are presumably less affected by CBD than by DPH. Although CBD and the other classical voltage-gated sodium channel blockers similarly target Na_v_1.2 channels [13], they are pharmacologically distinct anti-myotonia medications. Given the different burst intervals in the same patient, combining CBD with other voltage-gated sodium channel blockers (such as DPH, CBZ, and LTG) to treat myotonia in Na_v_1.4 muscle channelopathies may increase its effectiveness

### 4.3. Same Molecular Structure but Different Biological Functions in Cannabidiol and Tetrahydrocannabinol

In addition to CBD, tetrahydrocannabinol (THC) can also be extracted from hemp or cannabis. Both compounds interact with the endocannabinoid system; however, they have very different effects. The molecular structures for CBD and THC are the same, with 21 carbon atoms, 30 hydrogen atoms, and 2 oxygen atoms [12]. A slight difference in how the atoms are arranged accounts for the differing effects on the body. Both CBD and THC interact with cannabinoid receptors. The interaction affects the release of neurotransmitters in the brain which are responsible for relaying messages between cells and have roles in pain, immune function, stress, and sleep. Despite their similar chemical structures, CBD and THC do not have the same psychoactive effects [12,53]. CBD does not produce the high emotions associated with THC [12,53]. Nevertheless, CBD was effective in ameliorating anxiety, depression, and seizures. THC binds with the cannabinoid 1 (CB1) receptors in the brain. It produces a high or sense of euphoria. CBD can also bind to CB1 receptors but is very weak in binding. It needs THC for the binding and [13], in turn, CBD can help reduce some of the unwanted psychoactive effects of THC, such as euphoria or sedation. Given that CBD has a lower affinity for the endocannabinoid receptors than THC [13,54], several studies suggested that the anticonvulsant effects of THC and CBD in maximal electroshock and pilocarpine models occur with different mechanisms [13,50]. The THC activity is mostly through the CB1 receptor; however, the anticonvulsant effects of CBD are not [13]. These findings have inspired the growth of CB1- and CB2-independent focused research in epilepsy model.

## 5. Conclusions

CBD is able to relieve myotonia caused by sodium channelopathy, which results from high from high frequency repetitive discharge of the muscle membrane. Findings from the current study elucidate the binding kinetics of CBD onto the voltage-gated Na_v_1.4 channel of the muscle. CBD selectively binds to the Na_v_1.4 channel in its slow-inactivated state. The highly selective and efficient binding of CBD onto the slow-inactivated Na_v_1.4 channel provides novel therapeutic avenues to treat myotonia caused by sodium channelopathy.

## Figures and Tables

**Figure 1 biomedicines-09-01141-f001:**
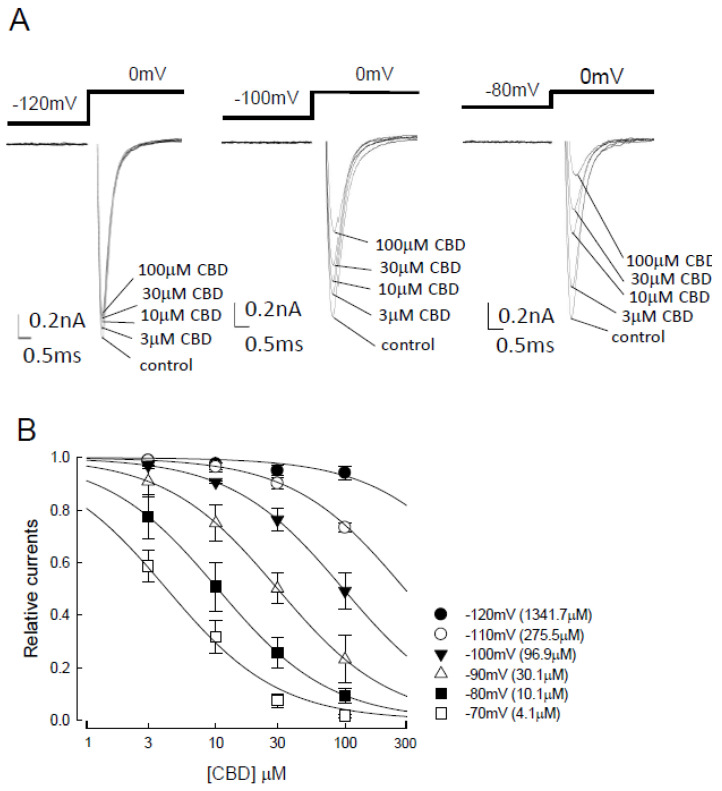
Inhibition of the activation of Na_v_1.4 sodium channel by cannabidiol at different holding potentials. (**A**) Current traces of the patched cells expressing Na_v_1.4 channels were recorded under treatment with different concentrations (0, 3, 10, 30, and 100 μM) of cannabidiol (CBD). The cells were held at −120, −100, and −80 mV, followed by depolarization to 0 mV for 5 ms. (**B**) Dose–response curves for inhibition of Na_v_1.4 sodium currents by CBD at different holding voltages between −120 and −70 mV (n = 5 for a low to high concentration of CBD). Peak currents in the presence of CBD were normalized to control peak currents at each of the different holding potentials and plotted against the CBD concentration. Lines are the fits of data using the following formula: relative current = 1/[1 + ([CBD]/K_app_)], where [CBD] was the CBD concentrations and K_app_ values were 1341.7, 275.5, 96.9, 30.1, 10.1, and 4.1 μM at the holding potentials of −120, −110, −100, −90, −80, and −70 mV, respectively.

**Figure 2 biomedicines-09-01141-f002:**
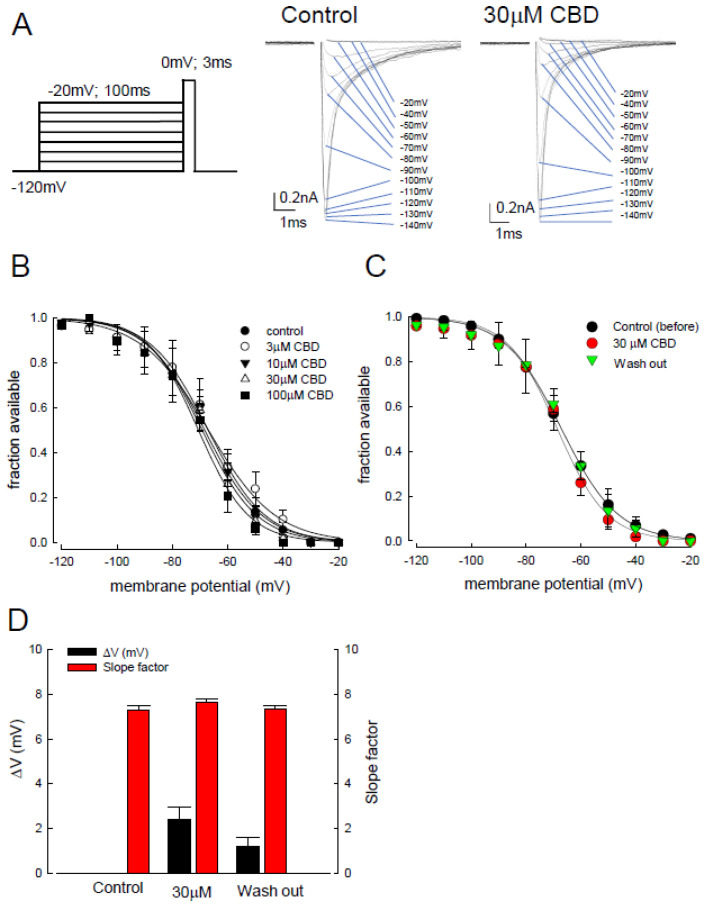
Cannabidiol induced changes in the fast inactivation curve of Na_v_1.4 channel. (**A**) In our electric protocol to test the fast inactivation of Na_v_1.4 channels, the cell was held at −120 mV and then stepped to the depolarizing pulse up to −20 mV (in 10 mV steps) for 100 ms, followed by a test pulse, 0 mV for 3 ms (left panel). The representative current sweeps of the Na_v_1.4 channel was recorded after the short test pulse (right panel). (**B**) The inactivation curve shifted by CBD from the cells expressing Na_v_1.4 channel is shown. Fraction available, defined as the normalized peak current (to the peak current with an inactivating pulse at −120 mV), is plotted against the voltage of the inactivating pulse, between −120 and −20 mV, to show the inactivating curve. The inactivating curve did not shift significantly under treatment with different CBD concentrations (3, 10, 30, or 100 μM CBD). (**C**) The inactivation curves of the control, CBD treatment, and washing out channels demonstrate no significant voltage shift during the approximately 100 ms depolarizing pulse. Lines are best fit using the Boltzmann function 1/(1 + exp[(V − V_h_)]/k), where V_h_ were −67.1, −71.7, and −67.4 mV, and k values were −10.4, −9.8, and −10.7 for control, those cells after treatment of 30 μM CBD, and those with washing out, respectively. (**D**) The shifts in inactivation curves (ΔV_h_) in Figure 2C were 1.8 and 1.5 mV for those cells with 30 μM of CBD and those after washing out, respectively. The k values were 7.3 ± 0.1, 7.6 ± 0.2, and 7.3 ± 0.2 for control, those cells with 30 μM of CBD, and those post washing out of CBD, respectively (n = 5).

**Figure 3 biomedicines-09-01141-f003:**
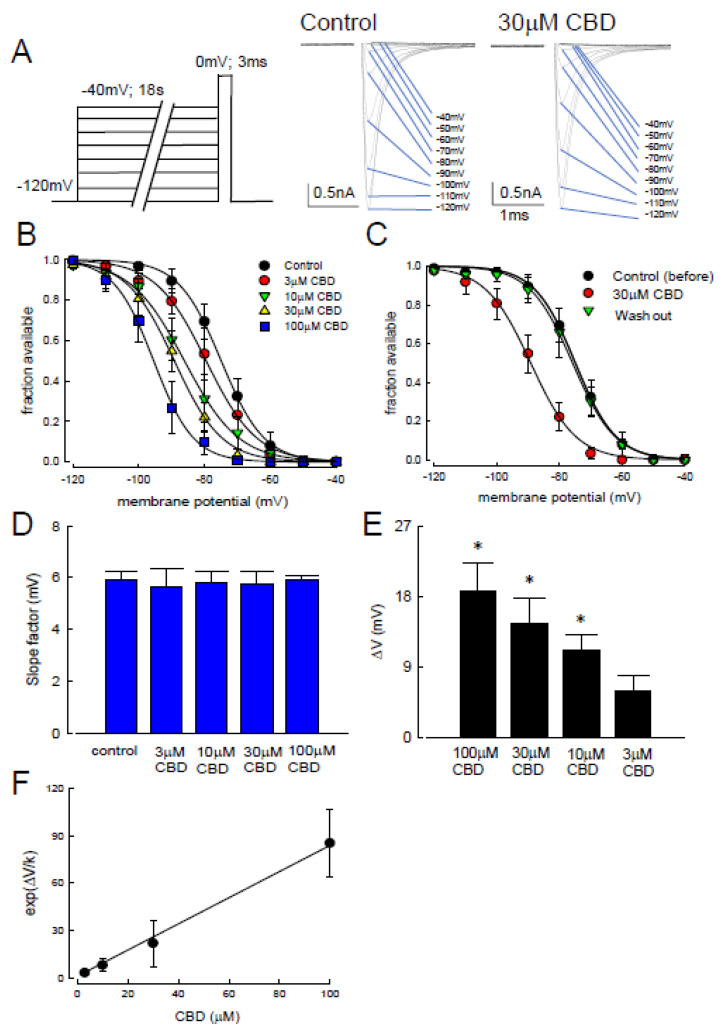
Changes in the slow-inactivating curve after the applications of cannabidiol. (**A**) The experiment protocols were similar to those of Figure 2, however, the duration of inactivation pulse was protracted to 18 s (left panel). The representative current sweeps for slow-inactivation of Na_v_1.4 in the control channels and those treated with 30 µM of CBD were recorded after the short test pulse (right panel). (**B**) The inactivating curve was shifted left in a dose-dependent manner after treatment with different concentrations of CBD. No significant change in slope was found in these curves. Lines of best fit were fitted using the Boltzmann function 1/(1 + exp[(V − V_h_)]/*k*), where V_h_ were −74.9 ± 0.4, −79.5 ± 0.6, −83.1 ± 0.5, −89.2 ± 0.6, and −95.5 ± 0.4 mV, and *k* values were −6.5 ± 0.3, −7.6 ± 0.5, −8.4 ± 0.4, −7.2 ± 0.4, and −6.0 ± 0.3 for the control channels and those channels treated with 3, 10, 30, and 100 μM of CBD, respectively. (**C**) The inactivation curves in the control channels, those treated with 30 μM of CBD, and those after washing out. There was a hyperpolarized shift of the inactivation curve in those channels upon treatment with 30 µM of CBD. Lines are fit using the Boltzmann function 1/(1 + exp [(V − V_h_)]/*k*), where V_h_ were −7.4 ± 0.4, −89.2 ± 0.6, and −75.9 ± 0.5 mV, and *k* values were −6.5 ± 0.3, −7.2 ± 0.45, and −6.8 ± 0.37 for the control channels, those with 30 μM CBD, and the washing out channels, respectively. (**D**) CBD did not significantly change the slope factor k. Cumulative results showed that the average values in the control channels and those channels after treatment with 3, 10, 30, and 100 μM of CBD were 5.8 ± 0.3, 5.6 ± 0.7, 5.8 ± 0.4, 5.7 ± 0.5, and 5.9 ± 0.1, respectively. (**E**) A dose-dependent shift in V_h_ for the inactivation curve. Comparing to the controls, the degree of voltage shift was 5.9 ± 1.9, 11.1 ± 1.9, 14.5 ± 3.3, and 18.7 ± 3.6 mV for those channels after treatment with 3, 10, 30, and 100 μM of CBD, respectively, (*, *p* < 0.05). (**F**) The exp(ΔV/k) values are plotted against the concentration of CBD. ΔV/k were derived from mean values Figure 3D,E. The line is a best fit to the data of the formula exp(ΔV/k) = 1 + ([CBD]/1.3), where the y-intercept is 1. The [CBD] denotes the CBD concentration in μM [42,43,44].

**Figure 4 biomedicines-09-01141-f004:**
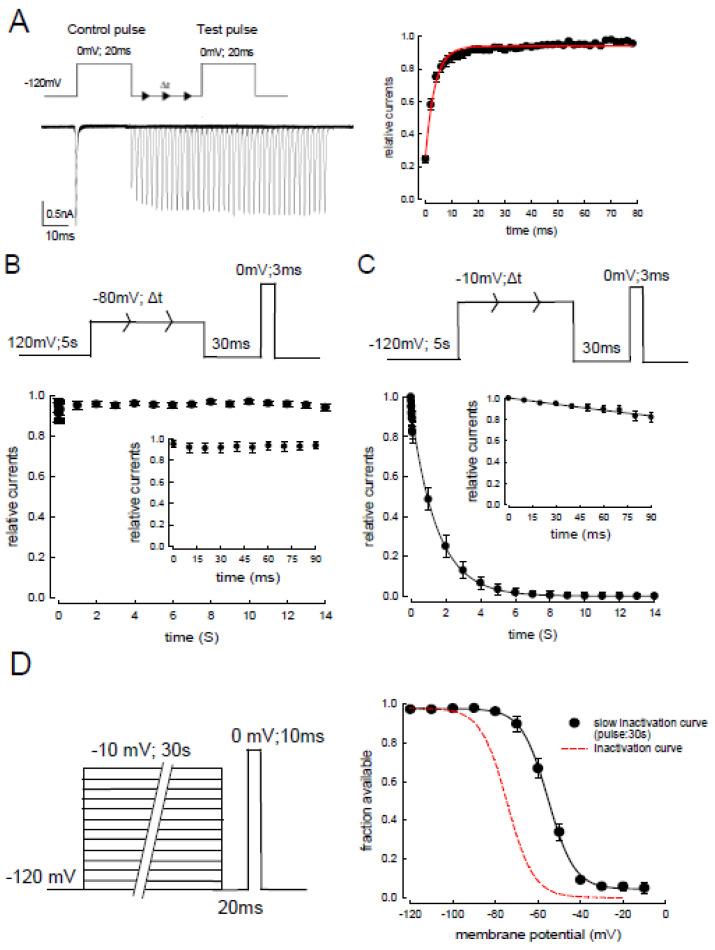
Recovery course from the fast and slow-inactivation curves in Na_v_1.4 channels. (**A**) The cell was held at −120 mV and pulsed twice to 0 mV (for 20 ms each) every 2 s, with a gradually lengthened gap between the two pulses at −120 mV. Sweeps were arranged so that the currents in the second pulse gradually shifted rightward as the gap lengthened (by 1 ms each sweep, left panel). The relative current, defined as the normalized peak current in the second pulse to that in the first pulse, is plotted against the duration of the intervening gap (n = 5, right panel). The plot shows that most channels entering the fast inactivation recovered after approximately 20 ms at −120 mV. (**B**) The electric protocol to evaluate the kinetics of slow-inactivation at −80 mV. The patch CHO-K1 cell was held at −120 mV for approximately 5 s, followed by a depolarized pulse −80 mV for a certain time. Then, the current of −120 mV for 30 ms was applied to the cell to allow full recovery from the fast inactivation. A test pulse at 0 mV for 3 ms was employed to evaluate the slow-inactivation. The elicited currents were normalized to the control current, relative currents, which was then plotted against the duration of the inactivating pulse at −80 mV (time in seconds). The inset figure is a close-up view of the first 90 ms of data. Most (>90%) of the Na_v_1.4 channels remained in the fast-inactivated state at −80 mV. (**C**) The electric protocol to evaluate the kinetics of slow-inactivation at −10 mV. The same protocols and analyses were employed as shown in Figure 4B, except that the inactivating pulse was set to −10 mV (n = 5). In this case, most Na_v_1.4 channels (>90%) entered the slow-inactivated state with a time constant of 1.43 s from the exponential fit to these points. (**D**) The patched cell was held at −120 mV, and the pulse protocol was repeated every 30 s. Depolarizing prepulses of varying voltages from −120 to −10 mV (approximately 30 s) were applied to the cells, followed by a gap voltage at −120 mV for 20 ms to allow recovery from fast inactivation. Subsequently, a test pulse at 0 mV for 10 ms was applied to evaluate the fraction of available channels. The curve was fitted using the Boltzmann function 1/1 + exp[(V − V_h_)/k], where the V_h_ value was −54.9 ± 0.7 mV and the k value was −7.3 ± 0.5 (the black curve, right panel). The left of the inactivation curve from Figure 3B is replotted with a dashed line on the voltage axis.

**Figure 5 biomedicines-09-01141-f005:**
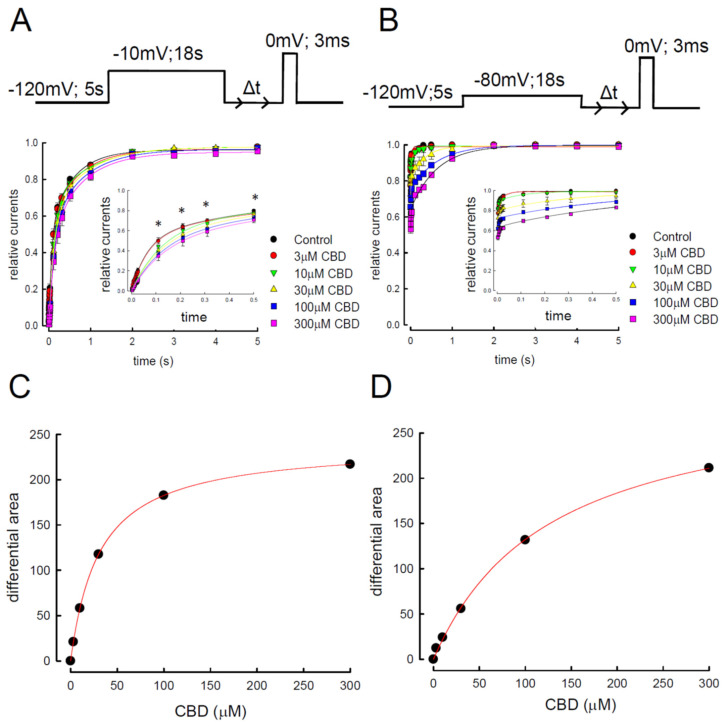
Recovery course from slow-inactivation and affinity of cannabidiol to the slow-inactivated Na_v_1.4 channel. (**A**) Patched cells expressing the Na_v_1.4 channel were held at −120 mV for 5 s. An inactivating pulse at −10 mV for 18 s was applied with a subsequent repolarization to −120 mV for a certain time to eliminate the channels with fast inactivation. The second test pulse of 0 mV was applied after an intervening gap at −120 mV for different lengths of time (0.003, 0.006, 0.009, 0.012, 0.015, 0.018, 0.021, 0.024, 0.027, 0.11, 0.12, 0.31, 0.51, 1, 2, 3, 4, and 5 s). Representative currents were overlaid according to the time sequence of examination. Time scales were applied for current kinetics only. The relative current, defined as the normalized peak current in the second test pulse (to the peak current in the first test pulse), was plotted against the length of intervening gap to obtain the time course of recovery from slow-inactivation in control (n = 5) Na_v_1.4 channels and those treated with different concentrations of CBD (n = 5). Note the close-up view of the first 500 ms of data shows a significant dose-dependent change in the recovery of slow-inactivation after CBD treatment (inset figure is a closed-up view of Figure 5A; * *p* < 0.05). (**B**) Approaches and analysis protocols were the same as in Figure 5A, except that the inactivating pulse was set at −80 mV for 18 s. (**C**) Difference between the area in control and upon different concentrations of CBD in Figure 5A to the maximal difference giving the relative difference in the area, which is plotted against the CBD concentration (see Appendix A). The line is the fit of data points using the following formula: relative difference in area = ([CBD]/31.5)/(1 + [CBD]/31.5), where [CBD] denotes CBD concentration in μM [41]. (**D**) Analysis protocols were the same as those of Figure 5B, except that the inactivating pulse was set at −80 mV for 18 s. The line is the fit using the following formula: relative difference in the area = ([CBD]/127.3)/(1 + [CBD]/127.3), where [CBD] denotes CBD concentration in μM [41].

**Figure 6 biomedicines-09-01141-f006:**
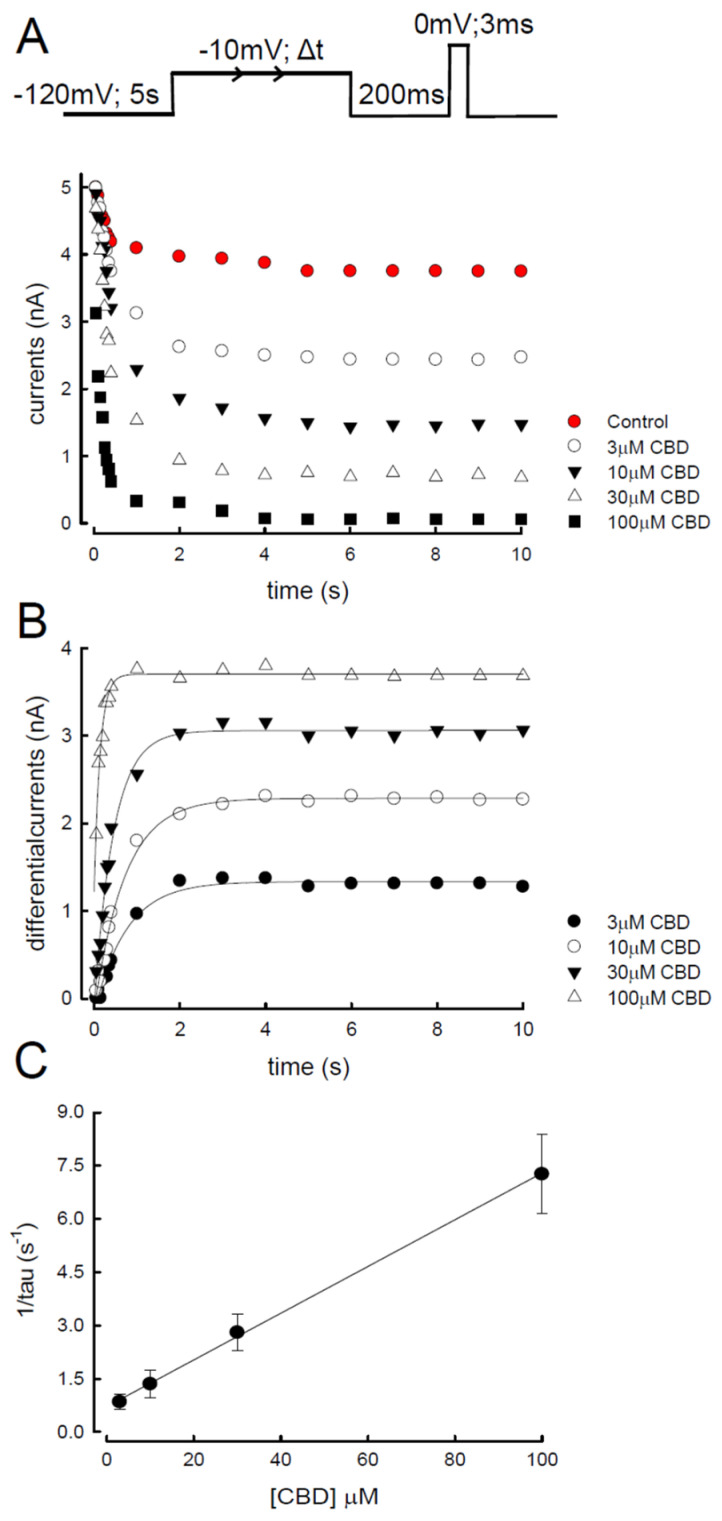
Binding rate of cannabidiol to the slow-inactivated Na_v_1.4 channel. (**A**) Patched cells were held at −120 mV for 5 s and stepped to an inactivating pulse at −10 mV for various durations of time (0.05, 0.1, 0.15, 0.2, 0.25, 0.3, 0.35, 0.4, 1, 2, 4, 5, 6, 7, 8, 9, and 10 s). The cells were then subjected to a fixed gap at −120 mV for 200 ms, which was set to allow for partial recovery of the slow-inactivated channels to obtain measurable currents elicited by the subsequent test pulse at 0 mV. Elicited currents were plotted against the duration of the inactivating pulse (−10 mV), and the representative currents were overlaid according to the time sequence. (**B**) Differential currents between the CBD and control conditions. Data were obtained from Figure 6A and are plotted against the duration of the prepulse. The time constants from the fit were 801.3, 740.2, 454.5, and 138.7 ms for the different CBD concentrations, namely, 3, 10, 30, and 100 μM, respectively. (**C**) The reciprocal of the time constant from that fitted in Figure 6B is plotted against the concentrations of CBD. The slope of the solid line is 64,000 M^−1^ s^−1^, and the y-intercept is 0.7 s^−1^.

**Figure 7 biomedicines-09-01141-f007:**
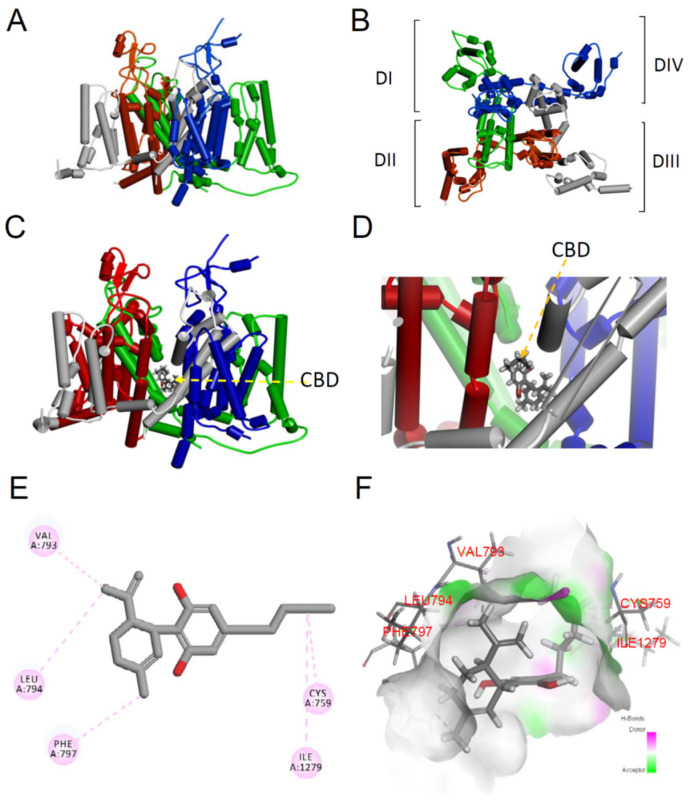
Homology modeling and docking interaction of the Na_v_1.4 channel with cannabidiol. (**A**) The homology model was built based on the X-ray crystal structures of human Na_v_1.4 channels (PDB: 6AGF) using the software Discovery Studio 2020 [37,38]. Four subunits of the Na_v_1.4 channels homology modeling is shown. (**B**) A regional view of the extracellular side of the pore in the Na_v_1.4 channel homology model, showing the transmembrane α-helix of the four domains. Domains I, II, III, and IV are colored in green, red, gray, and blue, respectively. (**C**) The 3D structure of the Na_v_1.4 channel with the CBD complex using molecular docking. CBD is shown as a CPK model. (**D**) Close-up view of the Na_v_1.4 channel with the CBD complex using molecular docking. (**E**) A 2D diagram showing the side chain of the Na_v_1.4 channel associated with the CBD molecule. Note that Val793, Leu794, Phe797, Cys759, and Ile1279 of the Na_v_1.4 channels are shown as the binding sites with CBD by the π alkyl interaction. (**F**) A 3D hydrogen bond surface plot at the binding site. The green color represents the conventional hydrogen bond, and the pink color represents the alkyl interaction.

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
