# Peer review of "Cannabidiol Selectively Binds to the Voltage-Gated Sodium Channel Nav1.4 in Its Slow-Inactivated State and Inhibits Sodium Current"

_biomedicines, 2021, doi:10.3390/biomedicines9091141_

Round 1

Reviewer 1 Report

Dear authors,

it has been my pleasure to read your interesting manuscript titled Cannabidiol inhibits sodium current via selectively binds to 2
the voltage-gated sodium channel Nav1.4 in its slow-inactivated 3
state, which is rationally structured and well-assembled overall. The methodology therein applied is solid and cogent. The tables and figures are significant and self-explanatory, thus providing a meaningful addition to the paper overall. As it is well known, the Nav1.4 channel, which is encoded by the SCN4A gene, is notably responsible for the depolarization of the skeletal muscle fibers. Numerous mutations in SCN4A have been linked to myotonic syndromes and periodic paralyses, highly debilitanting and disabling conditions. Gain-of-function variants in NaV1.4 has been associated with a wide-range of pathophysiological conditions such as inherited erythromelalgia, epilepsy, and arrhythmias. Given how therapeutic approaches for such conditions necessarily rely on a thorough understanding of the underlying mechanisms, the paper has merit in its thorough expounding upon cannabidiol, which unlike its closely-related psychoactive phytocannabinoid THC, has the necessary  hydroxyl moiety constituting an essential hydrogen-bonding interaction with the channel protein. That differentiation should probably be added in the manuscript.

In that regard, please consider citing the outstanding paper Sait LG, Sula A, Ghovanloo MR, Hollingworth D, Ruben PC, Wallace BA. Cannabidiol interactions with voltage-gated sodium channels. Elife. 2020 Oct 22;9:e58593. doi: 10.7554/eLife.58593. PMID: 33089780; PMCID: PMC7641581.

Overall, the article will make for a valuable contribution in a field of research with still much to be clarified, e.g.  the interactions of CBD and Navs and the structural underlying foundation on which such dynamics hinge. 

From a stylistic standpoint, I would recommend further review by a native English speaker to improve readability, although the article is quite well-written as it stands.

I believe the paper deserves publication in light of its significance, thoroughness and coherent structure.

Sincerely,

Author Response

Response to Reviewer 1 Comments

We thank the reviewer for the very helpful advice on the manuscript. We have carefully revised the manuscript according to the reviewer’s comments.

  1. it has been my pleasure to read your interesting manuscript titled Cannabidiol inhibits sodium current via selectively binds to the voltage-gated sodium channel Nav1.4 in its slow-inactivated state, which is rationally structured and well-assembled overall. The methodology therein applied is solid and cogent. The tables and figures are significant and self-explanatory, thus providing a meaningful addition to the paper overall. As it is well known, the Nav1.4 channel, which is encoded by the SCN4A gene, is notably responsible for the depolarization of the skeletal muscle fibers. Numerous mutations in SCN4A have been linked to myotonic syndromes and periodic paralyses, highly debilitanting and disabling conditions. Gain-of-function variants in NaV1.4 has been associated with a wide-range of pathophysiological conditions such as inherited erythromelalgia, epilepsy, and arrhythmias. Given how therapeutic approaches for such conditions necessarily rely on a thorough understanding of the underlying mechanisms, the paper has merit in its thorough expounding upon cannabidiol, which unlike its closely-related psychoactive phytocannabinoid THC, has the necessary hydroxyl moiety constituting an essential hydrogen-bonding interaction with the channel protein. That differentiation should probably be added in the manuscript.

Response:

According the reviewer`s suggestion, we have added the sentence in the revised manuscript [Page 15 to 16, Line 543 to 554].

In addition to CBD, tetrahydrocannabinol (THC) can also be extracted from hemp or cannabis. Both compounds interact with the endocannabinoid system, but have very different effects. The molecular structure for CBD and THC is the same with 21 carbon atoms, 30 hydrogen atoms, and 2 oxygen atoms [1]. A slight difference in how the atoms are arranged accounts for the differing effects on the body. Both CBD and THC interact with cannabinoid receptors. The interaction affects the release of neurotransmitters in the brain which are responsible for relaying messages between cells and have roles in pain, immune function, stress, and sleep. Despite their similar chemical structures, CBD and THC don`t have the same psychoactive effects [1,2]. CBD doesn`t produce the high emotion associated with THC [1,2]. Nevertheless, CBD was effective to ameliorate anxiety, depression, and seizures. THC binds with the cannabinoid 1 (CB1) receptors in the brain. It produces a high or sense of euphoria. CBD can also bind to CB1 receptors but is very weak in binding. It needs THC for binding and [3], in turn, CBD can help reduce some of the unwanted psychoactive effects of THC, such as euphoria or sedation. Given CBD has a lower affinity for the endocannabinoid receptors than THC [3,4], several studies suggested that the anticonvulsant effects of THC and CBD in maximal electroshock and pilocarpine models occur with different mechanisms [3,5]. The THC activity is mostly through the CB1 receptor, but the anticonvulsant effects of CBD are not [3]. These findings have inspired the growth of CB1- and CB2-independent focused research in the epilepsy model.

  1. In that regard, please consider citing the outstanding paper Sait LG, Sula A, Ghovanloo MR, Hollingworth D, Ruben PC, Wallace BA. Cannabidiol interactions with voltage-gated sodium channels. Elife. 2020 Oct 22;9:e58593. DOI: 10.7554/eLife.58593. PMID: 33089780; PMCID: PMC7641581.

Response:

We thank reviewer`s helpful comments and have added the outstanding paper in the revised manuscript [page 3, line 136, page 12; line 423, and page 14; line 480].

  1. From a stylistic standpoint, I would recommend further review by a native English speaker to improve readability, although the article is quite well-written as it stands.

RESPONSE:

We very thank reviewer`s very help comments. This manuscript has review by a native English speaker to improve readability. Certification is presented as the following sheet.

Reference

  1. Ghovanloo, M.R.; Abdelsayed, M.; Peters, C.H.; Ruben, P.C. A Mixed Periodic Paralysis & Myotonia Mutant, P1158S, Imparts pH-Sensitivity in Skeletal Muscle Voltage-gated Sodium Channels. Sci Rep 2018, 8, 6304, doi:10.1038/s41598-018-24719-y.
  2. Pertwee, R.G. The diverse CB1 and CB2 receptor pharmacology of three plant cannabinoids: delta9-tetrahydrocannabinol, cannabidiol and delta9-tetrahydrocannabivarin. British journal of pharmacology 2008, 153, 199-215, doi:10.1038/sj.bjp.0707442.
  3. Ghovanloo, M.R.; Shuart, N.G.; Mezeyova, J.; Dean, R.A.; Ruben, P.C.; Goodchild, S.J. Inhibitory effects of cannabidiol on voltage-dependent sodium currents. J Biol Chem 2018, 293, 16546-16558, doi:10.1074/jbc.RA118.004929.
  4. Straiker, A.; Dvorakova, M.; Zimmowitch, A.; Mackie, K. Cannabidiol Inhibits Endocannabinoid Signaling in Autaptic Hippocampal Neurons. Mol Pharmacol 2018, 94, 743-748, doi:10.1124/mol.118.111864.
  5. Devinsky, O.; Cilio, M.R.; Cross, H.; Fernandez-Ruiz, J.; French, J.; Hill, C.; Katz, R.; Di Marzo, V.; Jutras-Aswad, D.; Notcutt, W.G., et al. Cannabidiol: pharmacology and potential therapeutic role in epilepsy and other neuropsychiatric disorders. Epilepsia 2014, 55, 791-802, doi:10.1111/epi.12631.

Reviewer 2 Report

Review of the manuscript which has been submitted to Biomedicines

Manuscript no. biomedicines-1361327

Title:  Cannabidiol inhibits sodium current via selectively binds to the voltage-gated sodium channel Nav1.4 in its slow-inactivated state

  • The theme of the study is very well chosen and the electrochemical tests performed are complex and well thought out to follow exactly the effects of CBD on Na ion channels. The discussion section is well written and discussions of the results compared to the literature are relevant. Anyway, below I have made suggestions to improve the quality of the work.
  • I recommend changing the titles from 3.1 to 3.6 using the method that was used and not the conclusion that was obtained.
  • please rewrite or correct the English in the title
  • I kindly ask the author who wrote the paragraph between lines 61-76 to check English throughout the manuscript

Page 1 row 13 Cannabis is not the only plant that contains CBD. I recommend that bibliographic source https://doi.org/10.3390/antiox10050738, where you can identify valuable information for your study

Page 1 row 38 please reformulate “The DI and DII S4s are thought to be key for the activation of sodium channel activation,”

Page 1 rows 42 please reformulate “can be a pharmacological blocker interaction site”   

Page 2 rows 66   There are several ways to administer CBD, not just those presented by the authors

Page 3 rows 125 what external solution did you use?

Page 4 row 153 if the CBD concentration exceeds 60μM it becomes toxic to human fibroblasts (see https://doi.org/10.3390/antiox10050738) so higher concentrations than this would no longer be relevant for the study, if the authors think about a practical effectiveness of their study

Page 7 row 255 A lower ΔV value has a better or worse efficiency? Please detail this aspect for a better understanding (Fig 3E)

Page 15 row 533 I do not understand the connection between the first sentence of the conclusion and the rest of the conclusions.

Author Response

Response to Reviewer 2 Comments

We thank the reviewer for the very helpful advice on the manuscript. We have carefully revised the manuscript according to the reviewer’s comments.

  1. I recommend changing the titles from 3.1 to 3.6 using the method that was used and not the conclusion that was obtained.

Response:

We thank the reviewer`s comments and have to change the title from 3.1 to 3.6 as follows:

3.1 To assess the inhibition of transient sodium current by CBD with variable holding potentials on the CHO cells expressing Nav1.4 channel [page 4; lines 154-155]

3.2 The influence of CBD on the inactivated curve of Nav1.4 channels following a short depolarizing pulse [page 5; lines 194-195]

3.3 To explore whether CBD can bind to the slow-inactivated Nav1.4 channel with elongation of the depolarizing prepulse [page 6; lines 237-238]

3.4 The slow inactivated state of Nav1.4 channel occurred at the high depolarizing potential, -10 mV rather than at the low one, -80 mV [page 8; lines 288-289]

3.5 To evaluate the influence of different depolarizing potentials on the inhibitory effect of CBD [page 9; line 336]

3.6 To evaluate the inhibition of CBD by the time elapse for depolarization in Nav1.4 channel [page 11; line 390]

  1. Please rewrite or correct the English in the title

Response:

We have changed the title:

“Cannabidiol selectively binds to the voltage-gated sodium channel Nav1.4 in its slow-inactivated state and inhibits sodium current” [page 1; lines 5-6]

  1. I kindly ask the author who wrote the paragraph between lines 61-76 to check English throughout the manuscript

Response:

Many thanks for the comments! We have made some changes and correct the grammar errors in the paragraph which is stated in the following sentences.

“Recently, cannabidiol (CBD) was reported to be a potentially beneficial compound to treat epilepsy in children [1,2]. CBD is one of the 113 identified cannabinoids extracted from the cannabis plant [3]. Trials using CBD to treat a few disease entities such as anxiety, cognition decline, movement disorders, and pain had been conducted, although high-quality evidence to prove the effectiveness has not been obtained yet [4,5]. There are a few routes of CBD ingestion. It can be taken directly by mouth. The cannabis smoke can be inhaled by a nose. The aerosol spray of CBD can be absorbed by the cheek mucosa [6,7]” [page 2; lines 71-73].

Page 1 row 13 Cannabis is not the only plant that contains CBD. I recommend that bibliographic source https://doi.org/10.3390/antiox10050738, where you can identify valuable information for your study

Response

We thank the reviewer for the very kind reminder. We have written the sentence and cited reference which is recognized as an important reference to our future study [page 2; lines 69-71].

Page 1 row 38 please reformulate “The DI and DII S4s are thought to be key for the activation of sodium channel activation,”

Response:

Many thanks for the comments! We rephrase the sentence to “The segments S4s in the domains DI and DII are thought to be key molecules for the activation of the sodium channel, ….” [Page 1, lines 41-42].

Page 1 rows 42 please reformulate “can be a pharmacological blocker interaction site”

Response:

Thank you for the comment! We have rephrased the sentence to, “It has been reported that the pore of the Nav1.4 channel can be an interaction site for pharmacological blockers, such as lidocaine, benzocaine, and ranolazine [8,9]” [Page 1; lines 45 to page 2; line 46].

Page 2 rows 66 There are several ways to administer CBD, not just those presented by the authors

Response:

Thanks for the comment! The sentences have been changed as, “There are a few routes of CBD ingestion. It can be taken directly by mouth. The cannabis smoke can be inhaled by a nose. The aerosol spray of CBD can be absorbed by the cheek mucosa” [Page 2; lines 71-73].

Page 3 rows 125 what external solution did you use?

Response:

Thanks for the notification! We have added the details into the sentence, “the external solution which contains the following ingredient (mM): 150 NaCl, 10 HEPES, 2 CaCl2, and 2.5 MgCl2 (pH 7.6)” [Page 3; lines 122-123]

Page 4 row 153 if the CBD concentration exceeds 60μM it becomes toxic to human fibroblasts (see https://doi.org/10.3390/antiox10050738) so higher concentrations than this would no longer be relevant for the study, if the authors think about a practical effectiveness of their study

Response:

Many thanks for the comments! Upon the questions, we would raise a few arguments for our defense. Firstly, the study used the CHO-K1 cell for whole-cell recording. It seems that CHO-K1 cells is alive and still active in the electric activity when the concentration of CBD is high (~ 100 µM). In the excellent work from Petrovici et al [10], the toxic effect of CBD is tested in normal dermal fibroblast. The vulnerability in the primary cells might be rather different from the animal cell line, CHO-K1. Secondly, there is no energy failure evidence during the study. If the concentration is toxic to the cells, mitochondrial energy failure occurs at early stage which would perturb the cell membrane integrity resulting in aberration of sodium channel activation. We did not find any defects in the activation and inactivation curves of the patched sodium channel except the dose-dependent inhibition of sodium current by CBD. Finally, no significant apoptosis with changes in the medium during the experiment. Cell death can easily be detected in the culture dish. The process of apoptosis can change the pH values of the medium which would cause deviation of the activation and inactivation states. The experiments performed smoothly with the normal fitting curves. There are no significant changes in the slope factors at higher concentrations of CBD which argues against the occurrence of apoptosis of the patched cells.

Page 7 row 255 A lower ΔV value has a better or worse efficiency? Please detail this aspect for a better understanding (Fig 3E)

Response:

We are sorry for the obscure explanation about this issue. The ΔV is the voltage shift of the inactivation curve with different concentrations of CBD relative to the control. The hyperpolarization shift (left shift) means the inactivation occurs at a lower membrane potential. The larger value of ΔV, the more efficient the channel evolving into an inactivated state. Thus, if CBD binds to inactivated state of the channel, the higher concentration renders a larger value of ΔV [Page 7; lines 254-258].

Page 15 row 533 I do not understand the connection between the first sentence of the conclusion and the rest of the conclusions

Response:

Sorry to cause the misunderstanding! We have changed the sentences, “CBD is able to relieve myotonia caused by sodium channelopathy, which results from high from high-frequency repetitive discharge of muscle membrane” [Page 16; lines 567-568].

Reference

  1. Devinsky, O.; Marsh, E.; Friedman, D.; Thiele, E.; Laux, L.; Sullivan, J.; Miller, I.; Flamini, R.; Wilfong, A.; Filloux, F., et al. Cannabidiol in patients with treatment-resistant epilepsy: an open-label interventional trial. The Lancet. Neurology 2016, 15, 270-278, doi:10.1016/S1474-4422(15)00379-8.
  2. Patel, R.R.; Barbosa, C.; Brustovetsky, T.; Brustovetsky, N.; Cummins, T.R. Aberrant epilepsy-associated mutant Nav1.6 sodium channel activity can be targeted with cannabidiol. Brain 2016, 139, 2164-2181, doi:10.1093/brain/aww129.
  3. Campos, A.C.; Moreira, F.A.; Gomes, F.V.; Del Bel, E.A.; Guimaraes, F.S. Multiple mechanisms involved in the large-spectrum therapeutic potential of cannabidiol in psychiatric disorders. Philos Trans R Soc Lond B Biol Sci 2012, 367, 3364-3378, doi:10.1098/rstb.2011.0389.
  4. Black, N.; Stockings, E.; Campbell, G.; Tran, L.T.; Zagic, D.; Hall, W.D.; Farrell, M.; Degenhardt, L. Cannabinoids for the treatment of mental disorders and symptoms of mental disorders: a systematic review and meta-analysis. Lancet Psychiatry 2019, 6, 995-1010, doi:10.1016/S2215-0366(19)30401-8.
  5. VanDolah, H.J.; Bauer, B.A.; Mauck, K.F. Clinicians' Guide to Cannabidiol and Hemp Oils. Mayo Clin Proc 2019, 94, 1840-1851, doi:10.1016/j.mayocp.2019.01.003.
  6. Itin, C.; Barasch, D.; Domb, A.J.; Hoffman, A. Prolonged oral transmucosal delivery of highly lipophilic drug cannabidiol. Int J Pharm 2020, 581, 119276, doi:10.1016/j.ijpharm.2020.119276.
  7. Itin, C.; Domb, A.J.; Hoffman, A. A meta-opinion: cannabinoids delivered to oral mucosa by a spray for systemic absorption are rather ingested into gastro-intestinal tract: the influences of fed / fasting states. Expert Opin Drug Deliv 2019, 16, 1031-1035, doi:10.1080/17425247.2019.1653852.
  8. Lee, S.; Goodchild, S.J.; Ahern, C.A. Local anesthetic inhibition of a bacterial sodium channel. J Gen Physiol 2012, 139, 507-516, doi:10.1085/jgp.201210779.
  9. Gamal El-Din, T.M.; Lenaeus, M.J.; Zheng, N.; Catterall, W.A. Fenestrations control resting-state block of a voltage-gated sodium channel. Proc Natl Acad Sci U S A 2018, 115, 13111-13116, doi:10.1073/pnas.1814928115.
  10. Petrovici, A.R.; Simionescu, N.; Sandu, A.I.; Paraschiv, V.; Silion, M.; Pinteala, M. New Insights on Hemp Oil Enriched in Cannabidiol: Decarboxylation, Antioxidant Properties and In Vitro Anticancer Effect. Antioxidants (Basel) 2021, 10, doi:10.3390/antiox10050738.